# ZnO Nanostructures Doped with Various Chloride Ion Concentrations for Efficient Photocatalytic Degradation of Methylene Blue in Alkaline and Acidic Media

**DOI:** 10.3390/molecules27248726

**Published:** 2022-12-09

**Authors:** Razan A. Alshgari, Zaheer Ahmed Ujjan, Aqeel Ahmed Shah, Muhammad Ali Bhatti, Aneela Tahira, Nek Muhammad Shaikh, Susheel Kumar, Mazhar Hussain Ibupoto, Amal Elhawary, Ayman Nafady, Brigitte Vigolo, Zaffar Hussain Ibhupoto

**Affiliations:** 1Department of Chemistry, College of Science, King Saud University, Riyadh 11451, Saudi Arabia; 2Institute of Physics, University of Sindh Jamshoro, Jamshoro 76080, Pakistan; 3Department of Metallurgy, NED University of Engineering and Technology Karachi, Karachi 75270, Pakistan; 4Institute of Environmental Sciences, University of Sindh Jamshoro, Jamshoro 76080, Pakistan; 5Institute of Chemistry, Shah Abdul Latif University Khairpur Mirs, Khairpur Mirs 66020, Pakistan; 6Department of Zoology, Shah Abdul Latif University Khairpur Mirs, Khairpur Mirs 66020, Pakistan; 7Department of Chemistry and physics, Faculty of Education, Alexandria University, Alexandria 21500, Egypt; 8College of Natural Resources and Sciences, Jean Lamour Institute, Université de Lorraine, F-54000 Nancy, France; 9Institute of Chemistry, University of Sindh Jamshoro, Jamshoro 76080, Pakistan

**Keywords:** chloride (Cl^−^) concentration, ZnO, methylene blue, photodegradation

## Abstract

In this study, chloride (Cl^−^) ions were successfully doped into ZnO nanostructures by the solvothermal method. The effect of various Cl^−^ concentrations on the photocatalytic activity of ZnO towards the photodegradation of methylene blue (MB) under the illumination of ultraviolet light was studied. The as-prepared Cl^−^-doped ZnO nanostructures were analyzed in terms of morphology, structure, composition and optical properties. XRD data revealed an average crystallite size of 23 nm, and the XRD patterns were assigned to the wurtzite structure of ZnO even after doping with Cl^−^. Importantly, the optical band gap of various Cl ion-doped ZnO nanostructures was successively reduced from 3.42 to 3.16 eV. The photodegradation efficiency of various Cl^−^ ion-doped ZnO nanostructures was studied for MB in aqueous solution, and the relative performance of each Cl ion-doped ZnO sample was as follows: 20% Cl^−^-doped ZnO > 15% Cl^−^-doped ZnO > 10% Cl^−^-doped ZnO > 5% Cl^−^-doped ZnO > pristine ZnO. Furthermore, the correlation of the pH of the MB solution and each Cl ion dopant concentration was also investigated. The combined results of varying dopant levels and the effect of the pH of the MB solution on the photodegradation process verified the crucial role of Cl^−^ ions in activating the degradation kinetics of MB. Therefore, these newly developed photocatalysts could be considered as alternative materials for practical applications such as wastewater treatment.

## 1. Introduction

Various strategies have been employed to expel natural organic dyes from water contaminated due to leather products, pharmaceuticals, printing, food industries, modern coatings and plastics [1,2,3,4,5]. In view of the high molecular weight and synthetic steadiness of organic dyes, conventional methods such as physio-compounds and biochemicals normally do not offer the complete degradation of environmental contaminants [6]. Therefore, searching for new approaches for the removal of harmful dyes such as methylene blue (MB) from wastewater becomes highly demanded to maintain the aquatic life and environment. In recent years, a number of heterogeneous photosensitive materials have been applied for photodegradation and wastewater treatment [7], owing to marked advantages such as low cost, efficiency and stability under normal atmospheric conditions [8,9,10]. In this respect, zinc oxide (ZnO) is a semiconductor photocatalyst in group II–VI which possesses a large energy band gap of 3.37 (eV) and high exciton energy (60 m eV), along with being highly thermally and mechanically stable at room temperature [11,12,13,14,15]. Given the large band gap of ZnO, it requires high photonic energy for the photo-excitation process. Hence, photocatalytic activity cannot take place in some of the charge carriers due to the loss of energy during the recombination of electron–hole pairs. The pristine ZnO has a fast recombination of photogenerated electron–hole pairs, thereby resulting in the deterioration of the photocatalytic properties.

The prevention of a fast recombination rate with effective charge transport and separating efficiency of charge carriers could be achieved by introducing a dopant as an electron scavenger, thus enabling the material to increase the period of electrons in the excited state, and the absorption spectrum broadness could also be increased [16,17,18]. In view of these findings, the doping process is anticipated to contribute in three ways: (1) reducing band gap energy and enhancing adsorption, (2) decreasing resistivity and promoting the mobility of charge carriers and (3) varying the position of conduction and valence bands. In this context, ZnO as a semiconducting material was doped with other elements to reduce the fast recombination process and efficiently separate out the photogenerated electrons and holes [19,20,21,22]. Among approaches for ZnO doping, sol–gel [23,24,25], hydrothermal [26,27,28], chemical vapor deposition [29,30], semiconductor coupling [31,32], surface hybridization of ZnO with carbon [33] and combustion [34,35,36,37,38] have been implemented. A literature survey reveals that doping of ZnO with non-metals such as carbon (C), nitrogen (N) and sulfur (S) tunes the intermediate energy level band gap as well as the ability to absorb visible light, which leads to enhanced photocatalytic activity [39,40,41,42,43,44,45,46,47]. Moreover, Cl^−^-doped ZnO nanostructures have been reported for optical applications [48]. Despite the extensive studies on the topic of the design of photocatalysts, there are still more fundamental aspects which need attention, such as the effect of the pH of the dye solution and its correlation with each dopant level. Such understanding may lead to the development of photocatalytic materials for practical applications. Numerous studies have been carried out on the dopant effect on the photocatalytic applications; however, the role of each dopant concentration performance under acidic and alkaline conditions is rarely investigated. Hence, this study describes the role of different concentrations of Cl ion doping onto ZnO for the photodegradation of MB in acidic and alkaline conditions—there have not been any studies performed from this perspective in the available literature.

In this study, we used the solvothermal method for the synthesis of Cl ion-doped ZnO nanostructures and their application towards the photodegradation of MB under the irradiation of UV light. The hybrid material was studied with respect to structure, chemical composition and optical characterization. The synthesized 20% Cl^−^-doped ZnO showed remarkable photocatalytic efficiency of 99.94% for the degradation of MB in alkaline conditions.

## 2. Results and Discussion

### 2.1. Crystalline, Morphology and Composition Studied of Cl^−^-Doped ZnO Nanostructures

Figure 1a depicts the powder X-ray diffraction patterns of pristine ZnO and Cl^−^-doped ZnO nanostructures over the range of 23° to 78°. The various reflection peak positions and intensities of Cl^−^-doped ZnO were measured at 2Ө angles such as 31.86° (100), 34.54° (002), 36.32° (101), 47.62° (102), 56.66° (110), 62.96° (103), 66.44° (200), 68.04° (112) and 69.18° (201), which are fully supported by the JCPDS card No. 01-075-1533. The measured diffraction patterns of the prepared ZnO nanostructures were characterized by a hexagonal phase with a wurtzite structure [48]. This could be attributed to substitution of Zn ions by Cl^−^ ions from the lattice without disturbing the hexagonal structure of ZnO. However, a successive increase in the shift of two theta angle towards a higher angle with an increasing amount of dopant concentration was noticed, as shown in Figure 1b. The shift of two theta angle could be attributed to the change in lattice parameters due to successful doping of Cl ions. Furthermore, the obtained values of lattice parameters a and c and volumes of pure and Cl^−^-doped ZnO nanostructures were observed in increasing order, suggesting the incorporation of Cl^−^ atoms into ZnO, as shown in Figure 1c–e.

This increasing order may be justified owing to the larger ionic radius of Cl^−^ (r = 1.81Å) compared with O^2−^ (r = 1.40 Å), and this leads to the induction of consistent expansion of the ZnO lattice structures [49]. In addition, the doping of Cl^−^ into ZnO can further alter the distance in crystal structure as well as in diffraction angle. Moreover, the crystallite size of the as-prepared samples was evaluated by Scherer’s relation [50].
(1)$$D\left(nm\right)=\frac{K\lambda }{{\left(FWHM\right)}_{hkl}\;Cos\left(\theta \right)}\;$$
where *D* is crystalline size (nm), shape factor *K* = 0.94, *λ* is the wavelength of the X-ray (nm), β is the full width at half maximum (rad) and *θ* denotes the Bragg diffraction angle (deg). The average value of the crystalline size of the synthesized nanostructures was 23 nm, and their values were obtained in decreasing order with increasing Cl^−^ as a dopant concentration into ZnO. The decreasing order of crystallite size from 30.23 ± 0.04 to 13.66 ± 0.01 nm is most likely a contribution from the diffused and grown aspect of the ZnO crystal structure due to the marked difference in ionic radius between Zn^2+^ (0.74 Å) and Cl^−^ (1.8 Å). The chloride ion as a counter ion with other metal ions showed a reduction in the crystallite size due to a difference in the ionic radii of metallic ions and the chloride ion itself [51]. There is negligible effect on the d-spacing values, as given in Table 1. These size characteristics are essential parameters for possibly improving the photocatalytic properties of Cl^−^-doped ZnO nanostructures for potential applications in wastewater treatment as well as cleaning up the environment. These crystallographic values of the as-synthesized nanostructures are summarized in Table 1.

Scanning electron microscopy (SEM) analysis revealed that the addition of Cl^−^ ions as a dopant into ZnO significantly altered the morphology and crystal size from a nanorod to a thin film-like structure. As is evident in the Appendix A
Appendix A, the pristine ZnO has a short nanorod morphology with hexagonal facets. The addition of 5% Cl^−^ source as a dopant into the ZnO precursors resulted in almost the same nanorod-like morphology, but randomly aligned, as shown in Figure 2a. Further addition of 10% Cl^−^ dopant into the ZnO precursor yielded a slightly aggregated structure with the presence of a few nanorods (Figure 2b). Upon the addition of 15% Cl^−^ dopant, a compact cluster of nanorods was produced (Figure 2c). Finally, the addition of 20% Cl^−^ as a dopant into the ZnO precursor gave rise to a diffused and thin film-like morphology with some nanorod-like features, as shown in Figure 2d.

EDX was employed to quantitatively analyze the presence of zinc, oxygen and chlorine in the as-synthesized pristine ZnO and Cl^−^-doped ZnO samples, and the spectra are shown in Figure 3a–e. Closer inspection of the obtained EDX pattern emphasizes that undoped ZnO peaks appeared in the emission of X-rays, only related to zinc and oxygen atoms, as illustrated in the Appendix A
Appendix A. Figure 3a–d, obtained for 5%, 10%, 15% and 20% Cl^−^-doped ZnO, respectively, shows the presence of chlorine atoms along with zinc and oxygen atoms. Thus, the combined EDX data confirm the successful incorporation of Cl^−^ into ZnO nanostructures.

We explored the optical features of both pure ZnO and the four different samples of the as-prepared Cl^−^-doped ZnO nanostructures via UV–Vis spectroscopy. Figure 4 shows the absorbance spectra measured over the range of 200 to 600 nm. The absorbance values of each sample were in agreement with the reported works [1,25,27]. Careful analysis of the obtained spectra implied that increasing the Cl^−^ ion concentration by 5%, 10%, 15% and 20% causes the maximum wavelength absorption shift to a higher wavelength, which could be attributed to the agglomeration of the nanostructures, as reported elsewhere [15,52]. The energy gap conduction band (CB) and valence band (VB) may be assigned to the optical band gap, E. For hydrothermally synthesized photocatalysts, the Tauc equation is used to calculate band gap energy (E_g_):(2)$${\left(\alpha h\nu \right)}^{n}=k\left(h\nu -E_{g}\right)$$α denotes the absorption coefficient, K is a constant, hν is the photon energy, n = 2 and E_g_ (eV) represents the energy of the optical band gap.

Tauc plots of ZnO are shown in the Appendix A
Appendix A and Cl^−^-doped ZnO nanostructures are represented in Figure 4a–d. The insets in Figure 4a–d reveal the linear relationship between (αhν)^2^ and hν. This linear portion of the Tauc plot was extrapolated to the ordinate as equal to 3.08 eV. The band gap energies of Cl^−^-doped ZnO nanostructures were evaluated as 2.99, 2.88, 2.66 and 2.59 eV for 5%, 10%, 15% and 20% Cl^−^-doped ZnO, respectively. The measured optical band gap values were found according to the reported work [47]. The band gap energy follows slightly a decreasing order owing to the density of defects between the conduction band and valence band of Cl^−^-doped ZnO, which possibly arises from the Cl doping into ZnO. Thus, the non-metal (Cl)-doped ZnO causes a decrease in optical band gap energy, which indicates the potential of Cl^−^-doped ZnO nanostructures under UV light irradiation. The extrapolations in their insets represent 2.99, 2.88, 2.66, and 2.59 eV for 5%, 10%, 15% and 20% Cl^−^-doped ZnO, respectively.

### 2.2. Photodegradation Activity of Cl^−^-doped ZnO Nanostructures

The photodegradation of (MB) using pristine ZnO and Cl^−^-doped ZnO nanostructures was performed by irradiating UV light. The UV–Vis spectra of MB in aqueous solution at different pH values were recorded at different irradiation time intervals of 25 min for pure ZnO and 15 min for Cl^−^-ZnO, as presented in Figure 5a,b. Photodegradation activity of Cl^−^-doped ZnO nanostructures samples was inspected by pseudo first-order kinetics and linearly fitted graph Ln (C_o_/C_t_) versus time showing the degradation rate, as shown in Figure 5c,d, respectively.

The maximum absorbance peak in the spectra was measured at 664 nm. The decreasing height of the absorbance peak at 664 nm is a function of irradiation time, as shown in Figure 6 and Figure 7, thereby reflecting the degradation of MB dye at pH 5 and 11.

The illumination of UV light on the surface area of ZnO and Cl^−^-doped ZnO nanostructures causes the photogeneration of electron–hole pairs between conduction and valence bands, which is generally responsible for the ignition of the photocatalytic decomposition process leading to the degradation of dye pollutants. The photoexcited electron goes to CB, where it offers high oxidation potential to allow the direct oxidation of MB in the reaction medium. Furthermore, the photogenerated species in VB and in CB limit the recombination rate of electrons and holes by following the oxidation reaction with either H_2_O, OH^−^ or O adsorbed on ZnO and Cl^−^-doped ZnO nanostructures to produce the hydroxyl radicals through the following reactions [53,54]:(3)$$\mathrm{Cl}-\mathrm{ZnO}+\mathrm{hv}\;\to {\;e}_{\mathrm{CB}}^{-}+h_{\mathrm{VB}}^{+}$$
(4)$$e_{\mathrm{CB}}^{-}+O_{2\left(\mathrm{adsorbed}\right)\;}\to {\;O}_{2}^{•}\Rightarrow 2O_{2}^{•-}+{\;H}^{+}\to {\;H}_{2}O^{•-}+{\;O}_{2}^{•-}$$
(5)$$h_{\mathrm{VB}}^{+}+{\;H}_{2}O\to {\;H}^{+}+{\;\mathrm{HO}}^{•}$$
(6)$${\;e}_{\mathrm{CB}}^{-}+h_{\mathrm{VB}}^{+}\;\to \mathrm{Cl}-\mathrm{ZnO}+\mathrm{Heat}$$
(7)$$H_{2}O_{2}+e_{\mathrm{CB}}^{-}\;\to {\mathrm{HO}}^{•}+{\mathrm{OH}}^{-}$$
(8)$$H_{2}O_{2}+O_{2}^{•-}\;\to {\mathrm{HO}}^{•}+{\mathrm{OH}}^{-}+{\;O}_{2}$$
(9)$$H_{2}O_{2}+\mathrm{hv}\;\to 2{\mathrm{HO}}^{•}$$
(10)$$\mathrm{Dye}+\left(O_{2}^{•-}\;\to {\mathrm{HO}}^{•}+{\mathrm{HO}}_{2}^{•}\right)\to \mathrm{Intermediate}\to \mathrm{Product}\;$$

Finally, the hydroxyl radicals are used as scavengers to decompose the MB dye pollutant through the reactions represented in Figure 1 [54].

Figure 5a,b schematically depicts the slow decrease in the MB with a time of 200 min at pH 5 and 11, whereas in the case of Cl^−^-doped ZnO, the decrease in the height of the absorbance peak occurs much faster, with almost complete degradation of MB within an irradiation time of 90 min at pH 5 and 11, as illustrated in Figure 6a–d and Figure 7a–d, respectively.

The photocatalytic degradation activity may also be affected by various factors such as the amount of catalysts, the pH value and concentration of MB, the radiation of energy and the irradiation time. The effect of the concentration of Cl^−^ as a dopant into ZnO on the photodegradation of MB dye was investigated using different samples of Cl^−^-doped ZnO photocatalysts. For this study, the percentage of Cl^−^ as a dopant was varied from 5% to 20% of ZnO to optimize the performance of the photocatalysts. The 0.1 mM concentration of MB dye was kept constant for all samples to evaluate the effect of dopant concentration on the degradation process. The obtained data reveal that the photocatalytic degradation efficiency for MB with pristine ZnO was about 57.75% and 75.78% over 200 min at pH of 5 and 11, respectively, as shown in Figure 8a. The performance with 5%, 10%, 15% and 20% Cl^−^-doped ZnO over a period of 90 min was calculated to be 64.47%, 71.35%, 80.03% and 89.91%, respectively, at pH 5 (see Figure 8b). These values are markedly enhanced upon increasing the pH to 11, where under the same reaction time (90 min), degradation efficiencies of about 75.20%, 81.19%, 88.67% and 99.45% were achieved, as shown in Figure 8c. In view of these results, it is established that with a 20% Cl^−^ concentration in ZnO, the best photocatalytic activity was attained for MB degradation with efficiency levels of ∼89.91% and 99.47% within 90 min at pH values of 5 and 11, respectively, as represented in Figure 9b,c. This improved photocatalytic performance with 20% Cl^−^ dopant ZnO could be attributed to the presence of surface-active sites and a reduction in the optical band gap [55]. Thus, it can be concluded that 20% Cl^−^-doped ZnO gives the maximum degradation efficiency and hence may be considered as a promising photocatalyst for wastewater treatment.

It is widely accepted that the pH of a stock solution can be considered as a main factor and plays a vital role in driving the photosensitivity of a material through the surface charges [56,57]. For this purpose, we studied the effect of pH on the photosensitivity of as-prepared Cl^−^-doped ZnO materials towards the degradation of MB at pH 5 and 11. The pH value was set with the help of 0.1 M HCl and 0.1 M NaOH. The MB concentration of 0.1 mM was kept constant. In this study, we found that the photocatalytic efficiency of pure ZnO (without Cl^−^ doping) is ~50% and 55% at pH 5 and 11, respectively. The degradation efficiency of pristine n-type semiconductor ZnO was evaluated to be 57.75% and 75.78% over the time intervals of 200 min at pH 5 and 11, respectively, as shown in Figure 8a. Figure 8b shows that under same time and same conditions, the 20% Cl^−^-doped ZnO could degrade 89.91% at pH of 5, indicating that the degradation efficiency at pH 5 is less than at pH 11. On the other hand, with 20% Cl^−^-doped ZnO, the photocatalytic efficiency reached a maximum at pH of 11, with a degradation efficiency of 99.47% of MB over 90 min, as represented in Figure 9c. On the basis of this finding, it is evident that the degradation reaction is highly accelerated in alkaline media compared to acidic; this might be due to the greater electrostatic attraction between MB and the nanocomposites [58,59]. The minimum efficiency was observed at pH 5 and 11 for 5% Cl^−^-doped ZnO with degradation efficiency of 64.92% and 75.23%, respectively, as presented in Figure 8b,c. The obtained results of degradation efficiency are close to the previous studies.

The effect of pH on the photosensitivity of a material is complicated and could be understood through the related reported works [60]. Additionally, S. Sunitha et al. [61] and Iraj Kazeminezhad et al. [62] have proposed that both the acidic and basic features on the surface of the metal oxides as well as the density of zero-point charge (zpc) can be considered as the main factors in producing lower photocatalytic degradation performance at lower pH values and higher degradation efficiency at higher pH values.

The photocatalytic activity of pristine ZnO and Cl^−^-doped ZnO for the degradation of MB solution was evaluated under UV light. The evaluation in terms of quantifying the differences in degradation rates and obtaining the pseudo first-order kinetic constants was accomplished via using MB solution. Moreover, the rate of photocatalytic reaction and MB photo-decolorization rate constants were obtained using pseudo first-order kinetics, as shown in the following equation.
(11)$$Ln\frac{C_{o}}{C_{t}}=Kt$$
where *C_o_* is the concentration of MB before degradation, *C_t_* is the concentration of MB after degradation at time (*t*) and *k* is the pseudo first-order rate constant. The degradation rate coefficient was obtained through the linear fitting of Ln (*C_0_*/*C_t_*) as a function of time. The degradation rate was highly enhanced with the successive addition of Cl^−^ concentration into ZnO. Figure 4d shows the linearly fitted kinetic plot between $Ln\frac{C_{o}}{C_{t}}$ and time interval (min) yielding the degradation rates for pristine ZnO at pH 5 and 11. The linearly fitted kinetic plot follows the pseudo first-order kinetics model, and operating the slope may yield the degradation rate constants of 0.004 min^−1^ and 0.006 min^−1^ at pH 5 and 11, respectively, as shown in Figure 5d. In the case of Cl^−^-doped ZnO, better degradation rates for MB at pH 5 and 11 were attained. Figure 9b,d depicts the linearly fitted graph between $Ln\frac{C_{o}}{C_{t}}$ and time (min), revealing that the degradation rate constants functions of Cl^−^ ion concentration with respect to ZnO at pH 5 and 11, respectively.

As more Cl ions appear on the surface of ZnO, it causes a slower recombination rate, which ultimately leads to a faster degradation rate at both pH 5 and 11 [63]. Moreover, the doping of Cl^−^ into solid ZnO may increase the number of electron–hole pairs escaping to the surface of ZnO nanostructures and therefore enhance the photodegradation rate and efficiency of the Cl^−^-doped ZnO. The estimation parameters such as degradation rate, pH value, degradation efficiency, time, optical band gap and regression coefficient (R^2^) are listed in Table 2.

It is worth mentioning that the increase in the concentration of Cl^−^ as a dopant from 5% to 20% caused a gradual change in the degradation rate, from 0.010 to 0.021 min^−1^ at pH 5, whereas at pH 11, it increased the degradation rate from 0.012 to 0.041 min^−1^, as shown in Figure 8d.

The maximum photocatalytic activity was measured with 20% Cl^−^-doped ZnO at pH 5, with a degradation rate five times higher than that of pristine ZnO. On the other hand, with 20% Cl^−^-doped ZnO at pH 11, it was six times higher than that of pristine ZnO. Thus, the two approaches are reported as the cause of the improved photocatalytic performance. First, this might be attributed to the combinational effect of Cl^−^ and ZnO. It was noticed by the reduction in the size and optical band gap of prepared Cl^−^-doped ZnO samples, which consequently led to the suppression of the recombination of electron–hole pairs [46]. In other words, in a particular system, the rate of interfacial charge transfer may be increased by a decrease in particle size, which consequently increases the number of active surface sites. In this case, hydroxyl and superoxide radicals are formed when the photogenerated charge carriers react with absorbed molecules. However, our findings revealed that the decrease in the particle size might not solely be responsible for the improved photocatalytic activity. The rate of degradation of MB might be affected by the amplification in the rate of the recombination process; thus, sufficiently small particles proportionally raise the activity [64]. In addition, the degradation rates for Cl^−^-doped ZnO follow an order of 20% Cl^−^-doped ZnO > 15% Cl^−^-doped ZnO > 10% Cl^−^-doped ZnO > 5% Cl^−^-doped ZnO at both pH values of 5 and 11.

Additionally, pH 11 yields the optimum photodegradation activity, giving rise to a degradation rate of 0.041 min^−1^. Significantly, the maximum values at pH 5 and 11 are in agreement with the reported literature, and the obtained results are justified with the dynamic approach in which increasing the pH of the medium may cause the creation of more available active sites, thereby yielding more hydroxyl radicals onto the catalyst surface [64]. Additionally, the decreasing trend in band gap energy influences the photocatalytic performance for waste water treatment. Our findings regarding band gap appeared in decreasing order, from 3.54 to 3.16, with increasing Cl^−^ concentration into ZnO. The band gap findings suggest that a decrease in the band gap energy can produce enhanced performance of Cl^−^-doped ZnO at both pH 5 and 11. The performance obtained with 20% Cl^−^-doped ZnO was 89.91% and 99.45% at pH 5 and 11, respectively, as shown in Figure 8b,c. The scavenger study was performed to find out the nature of radicals which are effectively involved in the degradation process, as shown in Figure 10. For the photodegradation of organic dyes, common radicals are considered, such as superoxide radical ions (^•^O^−2^), hydroxyl radicals (^•^OH) and photogenerated holes (h^+^). For this purpose, we used ascorbic acid, sodium borohydride and ethylenediamine tetraacetate (EDTA) for the evaluation of the role of (^•^O^−2^), as supported by previous studies [52]. The suppression of the degradation process is highly dominated by ascorbic acid, as shown in Figure 10.

Significantly, the prepared Cl^−^-doped ZnO nanostructures in this investigation showed much greater efficiency in the degradation process of MB upon exposure to UV light than previous studies, as summarized in Table 3. The degradation efficiency of the synthesized pristine ZnO and Cl^−^-doped ZnO for the removal of MB at pH 5 and 11 is as follows: ZnO < 5% Cl^−^-doped ZnO < 10% Cl^−^-doped ZnO < 15% Cl^−^-doped ZnO < 20% Cl^−^-doped ZnO. Thus, the 20% Cl^−^-doped ZnO might be considered as the optimum value for promising photocatalysts [65]. Its degradation efficiency at pH 5 and 11 was compared with the published works [66,67,68,69,70,71,72,73,74,75], as given in Table 3. It is obvious that the Cl^−^-doped ZnO material has either equal or superior degradation efficiency in highly alkaline conditions. The possible reason for the higher performance in a dye solution with high alkaline pH could be related to the high density of hydroxyl groups, which potentially improved the degradation reaction kinetics of MB in aqueous solution, and consequently, higher degradation efficiency was observed.

## 3. Materials and Methods

### 3.1. Used Chemicals

Zinc acetate-dihydrate (ZnC_4_H_6_O_4_), 25% aqueous ammonia solution (Merck), methylene blue organic dye (C_16_H_18_ClN_3_S) and ethanol (C_2_H_5_OH, 99.5%) were provided by Sigma-Aldrich, Karachi, Pakistan, and used without further purification. The methylene blue was dissolved in deionized water and material synthesis was also performed in the deionized water. The chemical structure of methylene blue (MB) is shown in Figure 11.

### 3.2. Preparation of Various Cl^−^-Doped ZnO Nanostructures by Solvothermal Method

The preparation of pristine ZnO and Cl^−^-doped ZnO was performed via the solvothermal method by mixing zinc acetate di-hydrate (ZnC_4_H_6_O_4_) and 25% ammonia as precursors for the fabrication of ZnO. Different concentrations of ammonium chloride (% *w*.*w*), i.e., 5%, 10%, 15% and 20%, were mixed in 5 mL of ethanol in four beakers, and an ultrasonic bath was employed for 30 min to achieve sufficient dispersion. Then, 2.22 g of zinc acetate di-hydrate was dissolved by slowly adding 5 mL of 25% ammonia solution. After that the samples were labeled as 5% Cl^−^-doped ZnO, 10% Cl^−^-doped ZnO, 15% Cl^−^-doped ZnO and 20% Cl^−^-doped ZnO for identification. To avoid the evaporation of aqueous solution, the beakers were sealed with aluminum foil. The Cl^−^-doped ZnO nanostructures were grown in a muffle furnace at 90 °C for 5 h. Finally, the four synthesized Cl^−^-doped ZnO nanostructured samples were collected via a filtration and drying process. The pristine ZnO material was obtained by the same process without the use of Cl^−^ source. A low-resolution scanning electron microscope was used at an accelerating voltage of 20 k for the evaluation of the morphology of the prepared materials, and elemental analysis was performed using energy-dispersive X-ray spectroscopy equipped with a SEM instrument. A UV–visible spectrophotometer was applied to record the optical features of the prepared materials with an absorption wavelength range of 200 to 700 nm. The crystalline structure of the obtained ZnO samples was investigated by the powder X-ray diffraction technique, consisting of X-rays from Cu anodes with the potential of 45 kV and a current of 45 mA. The synthesis of the prepared materials is shown in Figure 2.

### 3.3. The Evaluation of Photodegradation Effectiveness of Various Cl^−^-Doped ZnO Nanostructures

The optical band gap calculations were performed with the use of 2 mg of each ZnO sample dispersed in 5 mL of deionized water, followed by sonication for 15 min. Afterwards, the significant dispersion of ZnO particles into deionized water and the settlement of solid particles at the bottom of the beaker occurred. Then, the top aliquot was placed in a cuvette cell and directly used for the measurement of the absorbance value. All the optical characterizations were performed with the suspension of various samples for the absorbance measurements. The effect of pH value on the photocatalytic activity of the as-synthesized pristine ZnO and Cl^−^-doped ZnO was examined in aqueous media. The pH of the MB aqueous solution was adjusted with the use of 0.2 M NaOH and 0.2M HCl. The reaction suspension was prepared using 10 mg of Cl^−^-doped ZnO powder in 100 mL of 0.0001M MB. The aqueous suspension of nanostructures was placed in a sonication bath for 30 min, which resulted in adsorption–desorption equilibrium. The UV–visible light irradiation was carried out in a locally made photoreactor using a compact UV lamp of 365 nm with a power of 18 watts. A UV–visible spectrophotometer (PE Lamda35) was employed to quantify the absorbance peak of MB dye, which was detected at 664 nm. After every 15 min of UV–Vis irradiation, the amount of MB was monitored by sampling out 5 mL of aqueous solution. The degradation of MB was observed at the wavelength range of 200–800 nm. The mathematical formula for the evaluation of the degradation activity of pristine ZnO and Cl^−^-doped ZnO samples was as follows:(12)$$\mathrm{Degradation}\mathrm{Efficiency}\;(\%)=\left(\frac{A_{o}-A}{A_{o}}\right)\times 100$$
where $A_{o}$ is the absorbance of the fresh dye solution without photo treatment (mg/L), and *A* represents the absorbance of MB dye (mg/L) at various irradiating times of UV light.

## 4. Conclusions 

We successfully synthesized four new Cl^−^-doped ZnO samples having 5%, 10%, 15% and 20% concentrations of Cl^−^ ions. We compared its photocatalytic degradation efficiency towards MB in acidic (pH = 5) and basic (pH = 11) media to that of pristine ZnO. The effect of pH on the degradation of MB as well as the role of different concentrations of Cl^−^ as a dopant into ZnO was investigated. The obtained results for the band gap energy of all Cl^−^-doped samples were found in a decreasing trend, whereas the efficiency of the photocatalytic degradation of MB was observed in a increasing trend. The doping of Cl^−^ ions induced the reduction in the optical band gap of ZnO, created defects in the crystal structure and consequently enhanced photocatalytic activity towards the degradation of MB in alkaline conditions. The obtained maximum degradation constants for the 20% Cl^−^-doped ZnO were (0.021 min^−1^) and (0.041 min^−1^), five and six times larger than that of pristine ZnO, at pH 5 and 11, respectively. Upon irradiation for ~200 min, the degradation efficiency of pristine ZnO was 57.75% and 75.78% at pH 5 and 11, respectively, whereas much higher values of 89.91% and 99.45% were obtained for the 20% Cl^−^-doped ZnO after irradiating with UV light for only 90 min. The photocatalytic efficiency upon using Cl^−^-doped ZnO nanostructures for the degradation of MB was observed in the order of 20% Cl^−^-doped ZnO > 15% Cl^−^-doped ZnO > 10% Cl^−^-doped ZnO > 5% Cl^−^-doped ZnO > pristine ZnO. The obtained results confirm that the Cl^−^-doped ZnO nanostructures may be considered as promising and alternative photocatalysts for wastewater treatment and other related environmental applications.

## Data Availability

All data is included in the manuscript and in Supplementary Materials and there are no additional available.

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
