# Peer review of "ZnO Nanostructures Doped with Various Chloride Ion Concentrations for Efficient Photocatalytic Degradation of Methylene Blue in Alkaline and Acidic Media"

_molecules, 2022, doi:10.3390/molecules27248726_

Round 1

Reviewer 1 Report

The effect of Cl- concentration on the photocatalytic activity of ZnO towards the degradation of methylene blue (MB) is studied. The as-prepared Cl--doped ZnO nanostructures were analyzed in terms of morphology, structure, composition and optical properties. The obtained results are promising for use of the material as photocatalysts in wastewater treatment. However, the current shape of the manuscript need modification considering correction of some editing mistakes and recalculations.

Here are some comments for authors:

Some typos, and editing mistakes need to be corrected, i.e. in Figure 2. a). Pristine ZnO   not prestine.

In Table 1

-        reconsider Calculation of  the volume one more time as some values seemingly incorrect.

-        Indicate the (hkl) planes that you used for calculating the lattice parameters and d-spacing.

-        Explain why the d-spacing increases after increasing the Cl doping since some peaks in XRD are positively shifted after the doping.

Author Response

Reviewer 1 Cl

Comments and Suggestions for Authors

The effect of Cl- concentration on the photocatalytic activity of ZnO towards the degradation of methylene blue (MB) is studied. The as-prepared Cl--doped ZnO nanostructures were analyzed in terms of morphology, structure, composition and optical properties. The obtained results are promising for use of the material as photocatalysts in wastewater treatment. However, the current shape of the manuscript need modification considering correction of some editing mistakes and recalculations.

We are thankful to the reviewer, for useful comments and suggestions in improving quality of manuscript prior to publication

Here are some comments for authors:

Some typos, and editing mistakes need to be corrected, i.e. in Figure 2. a). Pristine ZnO   not prestine.

Ans. In the revised manuscript, we have corrected the silly mistakes.

In Table 1

-        reconsider Calculation of  the volume one more time as some values seemingly incorrect.

Ans. In the revised manuscript, we have updated the calculated values, however the values of volume were found according to reported works elsewhere.

-        Indicate the (hkl) planes that you used for calculating the lattice parameters and d-spacing.

Ans. In the revised manuscript, hkl are highlighted in the text

-        Explain why the d-spacing increases after increasing the Cl doping since some peaks in XRD are positively shifted after the doping.

Ans. This has been explained in the revised manuscript and it is obvious the increase in d spacing is very little and it could be connected human handling error. And we passed the comment on it in the revised draft

Reviewer 2 Report

The authors presented the synthesis and characterization of ZnO nanostructures doped with Cl- ions with various concentration, and they tested the photocatalytic activity of these prepared materials for the photodegradation of an organic dye. 

The article is well written, but it's level of novelty and significance is pretty low, considering the amount of published literature on photodegradation of organic dyes. 

However, I think that the paper can be published after some major revision. See comments below:

1. Line 130, the authors write: “However, an increasing order in the red shift in peak position plane (102) at the angle of (47.476° ) in both pristine ZnO and Cl- -doped ZnO samples was observed, which might alter the optical properties due to the difference in the size of Cl- and Zn atoms, as shown in Figure 2(b).”

It is unclear here why the authors mention a “red shift” while talking about an XRD pattern. Also in the sentence concepts about optical properties are mixed with concepts related to crystallinity and XRD pattern. The sentence is very confusing and should be re-written.

2. Lines 149-151: It does not make sense to express crystallite sizes generated by Sherrer equation with that amount of significant figures, since they are indeed not significant. 23.356 nm should be expressed as 23 nm. Please correct Table 1. Moreover, the error associated with the measurement should be added.

3.Figures should be all prepared again, so that the style is in agreement. Size and type of character of each axes in the graphs should be always the same. The labels should be always the same, with same size and same color. In figure 3 labels are yellow, in Figure 2 labels are black. In Figure 4 labels are all squeezed due to the re-sizing of the figure. All these details should be careful addressed so that the Figure have a uniform style. The EDX results inside each figure is barely visible.

4.Line 183. The authors should convert the weight % into atomic % to see if these values indeed match their expected composition.

5.The UV-Vis spectra of all samples unfortunately do not make much sense. This is because absorbance values up to 4 were recorded, while absorbance should not exceed values of 1 – 1.5. Please consider that the absorbance is a logarithmic scale of the transmitted photons. If absorbance values of 4 were recorded, basically no photons were coming to the detector, and therefore a huge error was made, and therefore also background substraction has a lot of error. The measurements should be repeated with diluted samples, so that the higher absorbance value is around 1.

6.Also, the authors did not indicate in the experimental section how all the characterization of samples is carried out. Which instruments are used for SEM, XRD, UV-Vis. Moreover, how the UV-Vis measurements are done. Did they measure powders or suspensions? Did they measure inside an integrating sphere to account also for scattered light? Did you measure in transmission mode? All these details must be added. UV-Vis spectra in Figure 5 must be re-measured. For all samples indeed, for wavelength lower than 300 nm, it looks like the background subtraction failed.  

7.Line 384, authors write: “Furthermore, this observation may be also explained as that introducing of various concentration of Cl ions as dopant into ZnO nanostructures may lead to large plasmonic effect that produces significant impact on reaction kinetics and degradation efficiency.”

8.Why are the authors speaking about plasmonic effect? Is there any previously published work showing plasmonic effect on halogen doped oxide nanostructures?

Author Response

Reviewer 2 Cl

Comments and Suggestions for Authors

The authors presented the synthesis and characterization of ZnO nanostructures doped with Cl- ions with various concentration, and they tested the photocatalytic activity of these prepared materials for the photodegradation of an organic dye. 

The article is well written, but it's level of novelty and significance is pretty low, considering the amount of published literature on photodegradation of organic dyes. 

However, I think that the paper can be published after some major revision. See comments below:

We are thankful to the reviewer, for useful comments and suggestions in improving quality of manuscript prior to publication

1. Line 130, the authors write: “However, an increasing order in the red shift in peak position plane (102) at the angle of (47.476° ) in both pristine ZnO and Cl- -doped ZnO samples was observed, which might alter the optical properties due to the difference in the size of Cl- and Zn atoms, as shown in Figure 2(b).”

It is unclear here why the authors mention a “red shift” while talking about an XRD pattern. Also in the sentence concepts about optical properties are mixed with concepts related to crystallinity and XRD pattern. The sentence is very confusing and should be re-written.

Ans. This has been modified in the revised manuscript

2. Lines 149-151: It does not make sense to express crystallite sizes generated by Sherrer equation with that amount of significant figures, since they are indeed not significant. 23.356 nm should be expressed as 23 nm. Please correct Table 1. Moreover, the error associated with the measurement should be added.

Ans. These changes are corrected in the revised manuscript

3.Figures should be all prepared again, so that the style is in agreement. Size and type of character of each axes in the graphs should be always the same. The labels should be always the same, with same size and same color. In figure 3 labels are yellow, in Figure 2 labels are black. In Figure 4 labels are all squeezed due to the re-sizing of the figure. All these details should be careful addressed so that the Figure have a uniform style. The EDX results inside each figure is barely visible.

Ans. In the revised manuscript, we have modified the figures and their quality is highly improved.

4.Line 183. The authors should convert the weight % into atomic % to see if these values indeed match their expected composition.

Ans. The inset of table in EDX spectrum is already gives the quantitative value of atomic% and it is more like appropriate to the added values of ammonium chloride for the doping of Cl ion. The most interesting we do share here that the EDX atomic % can vary form one part of sample to other part of sample with relatively low difference. Hence advanced techniques are more favorable to do get quantitative information of atomic% which we do not have in access at the moment. The interesting part is that we wanted to show the presence of dopant element along with other main elements of compound with some % of it in each sample. 

5.The UV-Vis spectra of all samples unfortunately do not make much sense. This is because absorbance values up to 4 were recorded, while absorbance should not exceed values of 1 – 1.5. Please consider that the absorbance is a logarithmic scale of the transmitted photons. If absorbance values of 4 were recorded, basically no photons were coming to the detector, and therefore a huge error was made, and therefore also background substraction has a lot of error. The measurements should be repeated with diluted samples, so that the higher absorbance value is around 1.

Ans. If reviewer is talking about eh Figure 5, then the absorbance is below 2 and the issue of higher absorbance is referred to the condition that we did not centrifuge the solution and separate out the ZnO particles and consequently we see higher absorbance.   Yes, we agree with the reviewer that on dilution, we can have low absorbance, however, the measured results are according to the reported works as indicated in the revised draft of manuscript.

6.Also, the authors did not indicate in the experimental section how all the characterization of samples is carried out. Which instruments are used for SEM, XRD, UV-Vis. Moreover, how the UV-Vis measurements are done. Did they measure powders or suspensions? Did they measure inside an integrating sphere to account also for scattered light? Did you measure in transmission mode? All these details must be added. UV-Vis spectra in Figure 5 must be re-measured. For all samples indeed, for wavelength lower than 300 nm, it looks like the background subtraction failed.  

Ans. In the revised manuscript, we have added the experimental conditions in detail. However, currently our system is out of order for premeasuring of data reported in the Figure 4, we apologize for this.

7.Line 384, authors write: “Furthermore, this observation may be also explained as that introducing of various concentration of Cl ions as dopant into ZnO nanostructures may lead to large plasmonic effect that produces significant impact on reaction kinetics and degradation efficiency.”

Ans. These corrections are made in the revised version of manuscript.

8.Why are the authors speaking about plasmonic effect? Is there any previously published work showing plasmonic effect on halogen doped oxide nanostructures?

Ans. This has been revised and modified during the revision of manuscript

Reviewer 3 Report

ZnO nanostructures doped with various chloride ion concentrations for efficient photocatalytic degradation of methylene blue  in alkaline and acidic media

This manuscript cannot be published in journal Molecules at the present form and major revisions are necessary.

1-The necessity of this study should be clearly in the manuscript

2 - EDX analysis shows the presence of carbon. How the authors justify this?

3 - Authors should improve the quality of Figure 5

4- It is preferable to characterize all prepared photoctalysts by FTIR.

5- it is preferred to characterize the prepared composite material using XPS too investigate the interaction between Cl-  and ZnO.

6 - In section 3.2-page 9 line 246, the authors report that hydroxyl radicals are used as scavengers to decompose MB dye pollutant. What does this mean by "scavenger"? Please check about this sentence.

7-The authors should calculate the electronegativities for the ZnO and Cl-ZnO, and then calculated the CB and the VB of ZnO and Cl-ZnO respectively.

8- The photocatalytic performance of 20%Cl/ZnO for MB degradation in the presence of different scavengers under UV irradiation should be investigated.

9 - The authors should have added the mechanism of degradation of MB dye by Cl-ZnO.

10- Manuscripts should refer to and cite as much as possible from the last five years. Some high-quality literatures about photocatalyst and sustainability of water in recent years can be referenced and cited, such as :

https://doi.org/10.1007/s13369-022-06899-y;

https://doi.org/10.1016/j.seppur.2021.119399;  

https://doi.org/10.1016/j.surfin.2020.100611

Author Response

Reviewer 3 Cl

This manuscript cannot be published in journal Molecules at the present form and major revisions are necessary.

We are thankful to the reviewer, for useful comments and suggestions in improving quality of manuscript prior to publication

1-The necessity of this study should be clearly in the manuscript

Ans. In the revised manuscript, we have highlighted the importance of presented work.

2 - EDX analysis shows the presence of carbon. How the authors justify this?

Ans. In the revised manuscript we have added sentence connected to the carbon presence in EDX spectrum. This is because, we used the zinc acetate dihydrate and during the synthesis there is possibility of presence of carbon due to use of reducing agents like aqueous ammonia solution.

3 - Authors should improve the quality of Figure 5

Ans. In the revised manuscript, the quality of Figure 5 is improved.

4- It is preferable to characterize all prepared photoctalysts by FTIR.

Ans. We really understand the reviewer demand of FTIR, unfortunately we are unable to add this data in the revised draft due to unavailability of instrument. Therefore, we apologize for this inconvenience

5- it is preferred to characterize the prepared composite material using XPS too investigate the interaction between Cl-  and ZnO.

Ans. Yes, we do agree with the reviewer about the XPS demand, unfortunately, we do not have access to this instrument in the current time, hence apologize for this again.  Yes, it is true to know the quantitative value of Cl on the surface of ZnO and other things like defects in the structure  which could facilitate the final application of hybrid material.

6 - In section 3.2-page 9 line 246, the authors report that hydroxyl radicals are used as scavengers to decompose MB dye pollutant. What does this mean by "scavenger"? Please check about this sentence.

Ans. In the revised draft, we have defined the term and modified the draft in the discussion section

7-The authors should calculate the electronegativities for the ZnO and Cl-ZnO, and then calculated the CB and the VB of ZnO and Cl-ZnO respectively.

Ans. We do understand the reviewer comment on  it, however we do not expertise on it and we are still not 100% sure about eh location of Cl into Zn-O so it would be another part of study which requires new studies to study and bring the material towards practical applications.  Yes, Of course, it not only require the experimental data but at the same time theatrical data to describe authentically the driving force of hybrid material for photocatalytic applications.

8- The photocatalytic performance of 20%Cl/ZnO for MB degradation in the presence of different scavengers under UV irradiation should be investigated.

Ans. In the revised manuscript, we have provided the scavenger study for the sample.

9 - The authors should have added the mechanism of degradation of MB dye by Cl-ZnO.

Ans. In the revised manuscript, we have provided the general mechanism for photodegradtaion.

10- Manuscripts should refer to and cite as much as possible from the last five years. Some high-quality literatures about photocatalyst and sustainability of water in recent years can be referenced and cited, such as :

https://doi.org/10.1007/s13369-022-06899-y;

https://doi.org/10.1016/j.seppur.2021.119399;  

https://doi.org/10.1016/j.surfin.2020.100611

Ans. In the revised manuscript we have cited these works .

Reviewer 4 Report

Dear editor,

ZnO nanostructures doped with various chloride ion concentrations for efficient photocatalytic degradation of methylene blue in alkaline and acidic media is quite interesting but There are some misses understanding and errors that require some modifications in the paper. After the following corrections paper may be Acceptable for publication:

1.The Abstract should be revised carefully and need improvement.

2.The English and grammatical mistakes should be revise carefully and the monoculture need improvement.

3. In This part the authors should add some values and the different types of applications of ZnO  solar cells  also the Introduction part needs to be corrected and used updated references regarding application of this materials[owing to marked advantages 43 such as low cost, efficiency and stability under normal atmospheric conditions]:

Suggested references: https://doi.org/10.4995/Thesis/10251/160621

https://doi.org/10.1016/j.protcy.2016.05.078,

https://doi.org/10.1016/j.ijleo.2021.168283
4. The authors can improve the novelty of this work, please check well this paragraph.
 [In this contribution, we describe a facile procedure for doping various concentrations 72 of Clions into ZnO nanostructures using wet chemical method. Additionally, the effect 73 of various Cl- -doped ZnO on driving the degradation kinetics of MB is studied under 74 different pH value. The synthesized 20% Cl- -doped ZnO showed remarkable photocata- 75 lytic efficiency for degradation of MB at pH 11.]

5. Preparation of various Cl- -doped ZnO nanostructures by wet chemical method, the authors should add Shema of procedure

6. the intensity seems to be the same for all the samples in Figure 2. a) XRD patterns for pure and Cl- -doped ZnO nanostructures, which can be indexed as a 137 single phase of wurtzite ZnO structure using referencing code: (01-075-1533).  The authors can add some comments it.

7. the authors should the image sizes of Figure 3 in order to compare the doped and undoped ZnO.

8. Figure 4. (a) depicts the elemental composition of undoped ZnO and (b-e) shows ingredients in 5%, 186 10%, 15%, and 20% Cl- doped ZnO. The authors should improve the figure,it is not readable.

8.  the authors need to add references for the values calculated for Energy gaps and the band gap in the Figure 5 is not clear. Suggested references: https://doi.org/10.1016/j.ijleo.2022.168854

Author Response

Reviewer 4 Cl

ZnO nanostructures doped with various chloride ion concentrations for efficient photocatalytic degradation of methylene blue in alkaline and acidic media is quite interesting but There are some misses understanding and errors that require some modifications in the paper. After the following corrections paper may be Acceptable for publication:

We are thankful to the reviewer, for useful comments and suggestions in improving quality of manuscript prior to publication

1.The Abstract should be revised carefully and need improvement.

Ans. It has been revised during revision of manuscript

2.The English and grammatical mistakes should be revise carefully and the monoculture need improvement.

Ans. Thanks, and the language of paper is highly improved during the revision process.

3. In This part the authors should add some values and the different types of applications of ZnO  solar cells  also the Introduction part needs to be corrected and used updated references regarding application of this materials[owing to marked advantages 43 such as low cost, efficiency and stability under normal atmospheric conditions]:

Suggested references: https://doi.org/10.4995/Thesis/10251/160621

https://doi.org/10.1016/j.protcy.2016.05.078,

https://doi.org/10.1016/j.ijleo.2021.168283

Ans. These citations are added in the revised draft of manuscript
4. The authors can improve the novelty of this work, please check well this paragraph.
 [In this contribution, we describe a facile procedure for doping various concentrations 72 of Clions into ZnO nanostructures using wet chemical method. Additionally, the effect 73 of various Cl- -doped ZnO on driving the degradation kinetics of MB is studied under 74 different pH value. The synthesized 20% Cl- -doped ZnO showed remarkable photocata- 75 lytic efficiency for degradation of MB at pH 11.]

Ans. In the revised manuscript, these changes are done.

  1. Preparation of various Cl- -doped ZnO nanostructures by wet chemical method, the authors should add Shema of procedure

Ans. In the revised manuscript, we have provided in the synthesis scheme.

  1. the intensity seems to be the same for all the samples in Figure 2. a) XRD patterns for pure and Cl- -doped ZnO nanostructures, which can be indexed as a 137 single phase of wurtzite ZnO structure using referencing code: (01-075-1533).  The authors can add some comments it.

Ans. In the revised manuscript, we have elaborated the XRD section.  The intensity variation is very negligible therefore, we could not see as the image of Figure 2 is compressed.

  1. the authors should the image sizes of Figure 3 in order to compare the doped and undoped ZnO.

Ans. This has been revised during revision process.

  1. Figure 4. (a) depicts the elemental composition of undoped ZnO and (b-e) shows ingredients in 5%, 186 10%, 15%, and 20% Cl- doped ZnO. The authors should improve the figure,it is not readable.

Ans. In the revised manuscript, we have improved the quality of Figure 4

  1.  the authors need to add references for the values calculated for Energy gaps and the band gap in the Figure 5 is not clear. Suggested references: https://doi.org/10.1016/j.ijleo.2022.168854

Ans. In the revised manuscript, we have cited the suggested reference.

Round 2

Reviewer 2 Report

The authors have overall improved the paper, but I still feel that there are some unclear points:

In particular, the UV-Vis spectra in Figure 5 do not make much sense. The y axes is different in all of them. The absorbance reaches values of over 2, which means that there is a big error in the measurement. Indeed, the signal looks noisy between 350 to 400 nm. Also in all of them, there is a spike around 400 nm, that does not look normal - most likely a error in the background subtraction. 

Authors should repeat these measurements, making a dilution of the sample so that the absorbance never reach values over 1 after background subtraction.  

The figures were improved slightly but they are still not homogeneous. All x and y axes characters have different sizes. some of the text in the figures looks squeezed and distorted. Authors should really improve the presentation in general. 

Author Response

Comments and Suggestions for Authors

We are thankful to the reviewer for useful comments and suggestions and interest on our paper. We have tried to follow the comments where necessary and redid the UV-visible absorbance as per the comment, we believe it would be appreciated by the reviewer

The authors have overall improved the paper, but I still feel that there are some unclear points:

In particular, the UV-Vis spectra in Figure 5 do not make much sense. The y axes is different in all of them. The absorbance reaches values of over 2, which means that there is a big error in the measurement. Indeed, the signal looks noisy between 350 to 400 nm. Also in all of them, there is a spike around 400 nm, that does not look normal - most likely a error in the background subtraction.

Authors should repeat these measurements, making a dilution of the sample so that the absorbance never reach values over 1 after background subtraction.  

Ans. All the UV-Vis Spectra of Figure 5 are repeated according the comments. However, we tried to reach the low value of absorbance through dilution but could not succeed, but at the same time we have cited the articles for the illustration of high absorbance value. The presented absorbance values are fully supported our absorbance measurement. The y axis same difference due to possible variation in the dispersion of sample in each case.  At the same time, we have recalculated the optical band gap which are fully supported by the existing literature.

The figures were improved slightly but they are still not homogeneous. All x and y axes characters have different sizes. some of the text in the figures looks squeezed and distorted. Authors should really improve the presentation in general. 

Ans. We have reviewed the Figures quality particularly the Figure 5 and S1 which were improved.

Reviewer 3 Report

The revised version can be accepted for publication

Author Response

Thanks for considering our work.

Reviewer 4 Report

The authors addressed all the comments and the paper can be accepted in present form.

Author Response

Thanks for considering our work.